# Quantitative mapping of pseudouridines in bacterial RNA

Shikha Sharma[1], Brendan Woodworth[1], Bin Yang[1], Ning Duan[1], Mannuku Pheko[1], Niki Moutsopoulos[2] & Akintunde Emiola [1] ✉

RNA pseudouridylation is one of the most prevalent post-transcriptional modifications, occurring universally across all organisms. Although pseudouridines have been extensively studied in bacterial tRNAs and rRNAs, their presence and role in bacterial mRNA remain poorly characterized. Here, we used a bisulfite-based deep sequencing approach to provide a comprehensive and quantitative measurement of bacterial pseudouridines using *E. coli*, to provide proof of concept. We identified 1,954 high-confidence sites in 1,331 transcripts, which is 29 times above previous estimates and representing almost 30% of the transcriptome. Furthermore, pseudouridines were significantly associated with mRNA stability and enriched in transcripts associated with secondary metabolite production and adaptation to diverse environments. Finally, we mapped pseudouridines in oral microbiome samples of human subjects, demonstrating the broad applicability of our approach in complex microbiomes. This way, we observe that, although uridines are required for modification, mRNAs from GC-rich bacteria harbored more pseudouridine sites than AT-rich genomes in our dataset. Altogether, our work highlights the advantages of mapping bacterial pseudouridines and provides a tool to study posttranscription regulation in microbial communities.

Pseudouridylation is one of the most abundant nucleotide modifications present in all domains of life[1,2]. Pseudouridines (Ψs) are found in rRNA, tRNA, and other non-coding RNA where they enhance base-pairing, RNA stability, and influence translation fidelity[3,4]. Recent transcriptome-wide studies in humans have further identified Ψs in eukaryotic mRNAs where pseudouridylation is altered in response to stress, which suggests a regulatory role in eukaryotes[5].

In many bacteria species, such as *E. coli*, modification is carried out by eleven different pseudouridine synthase (PUS) enzymes — five of which specifically modify rRNA targets (RluB, RluC, RluD, RluE, and RsuA)[6,7]. On the other hand, TruA, TruB, TruC, and TruD explicitly modify tRNAs, while RluA and RluF pseudouridylate both rRNA and tRNA[6]. Although lack of Ψ in eukaryotic rRNA severely impacts ribosomal function, a mutant *E. coli* strain devoid of any Ψ in the ribosomal subunit did not show any major effect on growth, decoding and

ribosome biogenesis[7]. Among the tRNA PUS enzymes, TruA appears to be the most important. Mutations in TruA, which is responsible for pseudouridylation in anticodon loop, affects the translation machinery[8]. In particular, tRNA lacking Ψs in this loop, are unable to proceed at aminoacyl-tRNA transfer step due to lack of stability in the mRNA-tRNA complex[8].

Most in vivo studies investigating the impact of pseudouridylation in bacteria have been focused on rRNA and tRNA. This is partly due to the unavailability of experimental toolset to investigate the distribution and function of Ψ in mRNA. Previous in vitro studies have demonstrated that replacement of uridine with Ψ in bacterial mRNA can impede amino acid addition and increase the occurrence of amino acid substitutions[9]. Similarly, Ψ at stop codons (UAA, UAG, or UGA) can enable ribosomal readthrough of the modified stop codon[10]. Based on these observations, Ψs in mRNAs are believed to

[1]Microbial Therapeutics Unit, National Institute of Dental and Craniofacial Research, National Institutes of Health, Bethesda, MD, USA. [2]Human Barrier Immunity Section, Laboratory of Host Immunity and Microbiome, National Institute of Allergy and Infectious Diseases, National Institutes of Health, Bethesda, MD, USA. ✉e-mail: akintunde.emiola@nih.gov

influence protein translation in bacteria, but yet to be investigated in vivo.

There has been an attempt to identify pseudouridylation sites in bacterial mRNA at base-level resolution[11]. The authors relied on Pseudo-seq, a transcription-wide approach for Ψ profiling[12]. However, Pseudo-seq is known to suffer from low sensitivity and typically identifies very few pseudouridylation sites. In fact, newer methods that uses bisulfite (BS) to label Ψ (e.g., BID-seq[13,14] and PRAISE[15]) have recently been shown to identify ~23 times more Ψ sites than Pseudo-seq[15]. Therefore, new approaches are required to evaluate the assumed widespread distribution of Ψ in bacterial mRNAs.

Furthermore, studying mRNA pseudouridylation in microbiomes may help elucidate the role of a set of genes under a given condition. Typically, analysis of gene expression of the microbiota is performed using RNA-seq (i.e., metatranscriptomics). In this case, mRNA level in sequence data is used as an indirect proxy for protein abundance. Given that mRNA pseudouridylation can potentially influence translation[13], conventional RNA-seq analysis may not accurately recapitulate the protein translation landscape of the microbial community. Consequently, new methods capable of incorporating this widespread post-transcription modification will provide a more accurate representation of protein dynamics. To date, there is no study that has examined mRNA pseudouridylation or post-transcriptional regulation in microbiomes.

In this work, we adapted a BS-based approach to identify Ψ sites in *E. coli* RNA at base resolution and identified Ψ sites in one-third of mRNAs. We also establish that Ψ in protein-coding RNAs enhances transcript stability. Lastly, we demonstrate the applicability of our method in complex microbiome samples. Altogether, our work provides a tool to study post-transcription regulation in isolate bacteria and complex microbial communities.

## Results

### A BS-based approach to identify Ψ positions in bacterial RNA

BS-based approaches were recently developed to identify the positions of Ψs in human mRNA[13–15]. However, due to the instability of prokaryotic mRNAs and the absence of poly-A tails, these methods are not readily applicable to bacteria. In BS methods, treatment of mRNA with BS generates BS-Ψ adduct, which subsequently induces deletion at Ψ sites during reverse transcription (RT). By mapping these single-base deletions with untreated samples (which have no deletions), the exact location of Ψs can be identified (Fig. 1). We adapted the BS-based approach with modifications and applied it to *E. coli* samples.

We began by extracting total RNA and enriching for non-rRNAs through ribodepletion. Next, we performed fragmentation and split samples into two halves, with one half treated with BS (Fig. 1). After RT and sequencing, we mapped reads to the *E. coli* genome, realigned reads to increase sensitivity for reads that contain gaps[16] and retrieved the coverage of each nucleotide. Based on thresholds from previous

studies[13,15], we considered a uridine position as pseudouridylated if (i) the coverage depth in both BS-treated and untreated samples is ≥20; (ii) deletion rate is ≥5% in BS-treated samples but less than 1% in untreated samples; and (iii) deletion count is above 5 in BS-treated samples. Therefore, the deletion ratio is proportional to Ψ levels. For instance, a deletion ratio of 0.5 suggests 50% of transcripts harbor Ψ in a site.

### Validation of BS-based profiling of bacterial Ψs

To validate our approach, we used a wild-type (WT) *E. coli* and a mutant strain with knockout deletions in all seven rRNA PUS enzymes[7] (*rluA, -B, -C, -D, -E, -F*, and *rsuA*). This mutant is incapable of pseudouridylating 16S and 23S rRNA and subsequently referred to as Ψ^ΔrRNA strain. We cultured cells under optimal (37 °C) and various stress conditions (28 °C, ampicillin, gentamicin, and high NaCl) to maximize the detection of Ψ sites, especially in transcripts which are mostly expressed under sub-optimal conditions (Supplementary Fig. 1A). Using the pipeline described above, we initially sought to determine the distribution of all base deletions in our dataset. Because the GC content of *E. coli* is approximately 50% and assuming deletions occur randomly, the expected deletion counts for a given base, relative to other bases, is expected to be similar under non-treated conditions. In other words, the average deletion ratio for one base (e.g., U), relative to others should be ~1. (i.e., mean(U:A, U:G, U:C)). Indeed, in untreated samples, we observed the expected deletion frequencies for each base (Fig. 2A). In contrast, U-sites, but not other bases, were deleted >5 times above the expected frequency in BS-treated samples. This is consistent with the hypothesis that BS-treatment results in deletions at Ψ positions[13–15].

For downstream analysis, we included an additional filter to minimize the detection of false positives. We required a putative Ψ site to be identified in at least two samples — whether from biological replicates or different stress conditions. This resulted in 1954 high-confidence sites in 1331 transcripts, representing almost 30% of the transcriptome. Importantly, the deletion rates were highly reproducible across biological replicates, indicating a quantitative ability of BS-based methods to detect Ψ in bacterial mRNAs (Supplementary Fig. 1B, Supplementary Data 1). To further validate our approach, we searched for known Ψ sites in rRNA. Although all samples were subjected to ribodepletion during sample prep (Fig. 1), there is usually a small percentage (~0.5%) remaining, which is sufficient for many analyses[17]. Our pipeline accurately identified the sole Ψ site in 16S rRNA, and 8 of 10 sites in 23S rRNA (Fig. 2B, C). In contrast, we did not detect any Ψ in rRNAs from Ψ^ΔrRNA samples (Fig. 2B, C). Therefore, our approach accurately identified known Ψ positions in rRNA with no false positives detected.

Next, we sought to identify established Ψ sites in tRNAs. Pseudouridines are known to be deposited in position 13 (D-stem loop), positions 32, 38, 39, and 40 in the anticodon-stem loop, and positions

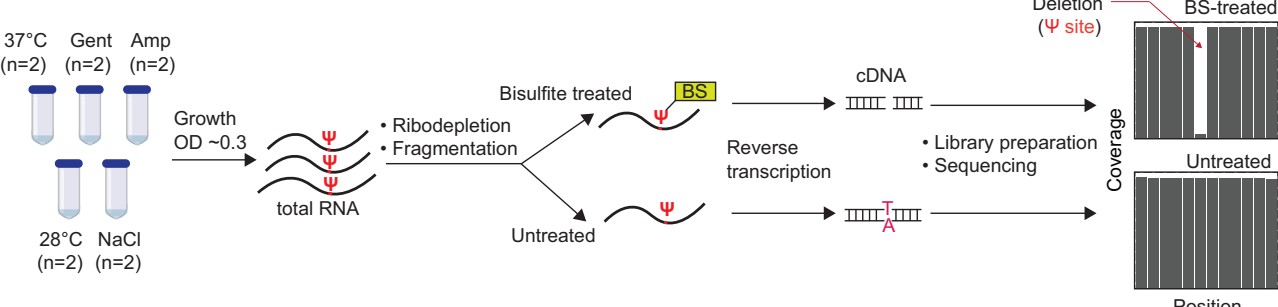

**Fig. 1 | Pipeline for detection of Ψ sites in bacteria RNA.** BS treatment of RNA induces deletion at Ψ locations during cDNA synthesis. After sequencing, deletion sites in BS-treated samples are compared with untreated samples to accurately identify Ψ positions. Gent gentamicin, Amp ampicillin.

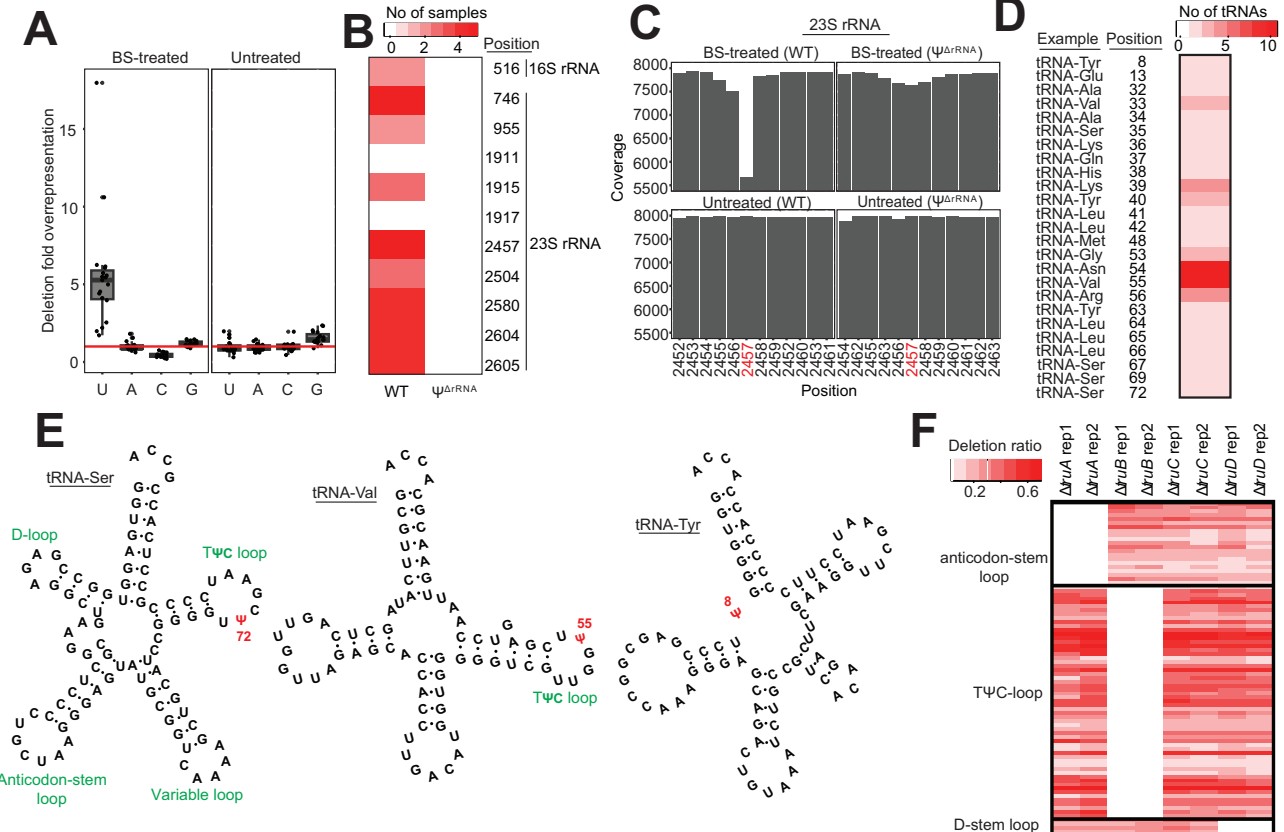

**Fig. 2 | Bisulfite-based profiling accurately detects known Ψ sites in bacterial rRNA and tRNA. A** Deletion frequency of each nucleotide under BS-treated and untreated conditions. Each data point represents the average per-sample deletion ratio for a base relative to others ($n = 19$). Since the GC content of *E. coli* is approximately 50%, random deletions, relative to other bases should be similar. The horizontal line represents the expected deletion frequency if deletion occurs randomly. BS-treatment induces deletions specifically at uridine sites. Center lines in boxplots represent the median and the edges represent the lower and upper quartiles. Whiskers show values that fall within 1.5× of the interquartile range. **B** Detection of known Ψ positions in rRNA from WT and a mutant strain unable to pseudouridylate rRNA (Ψ^ΔrRNA). The heatmap shows the number of samples where Ψ was detected (red = present; white = absent). **C** Representative view of a Ψ site (position 2457) in 23S rRNA from WT and Ψ^ΔrRNA strains. **D** Identification of Ψ sites in tRNAs. The heatmap shows the number of tRNAs with detected Ψ at a given site. Representative examples of tRNA harboring Ψ modification in a particular site are also provided. **E** Examples of tRNA structures (tRNA-Ser and tRNA-Val) harboring Ψ in exact locations in the TΨC loop despite having varying length. The structure on the right (tRNA-Tyr) shows a newly identified Ψ site in position 8. **F** Ψ proportion (deletion ratio) from biological replicates of tRNA PUS mutants. Each row represents a Ψ site in tRNAs (red = present; white = absent). Source data are provided as a Source Data file.

55 and 65 in the TΨC-loop[6,18,19]. Similarly to the rRNA data above, we accurately detected all known tRNA sites (Fig. 2D), despite differences in length due to the variable arm region. For instance, with our pipeline, we found position 72 in the TΨC loop of tRNA-Ser is pseudouridylated, which is equivalent to position 55 in the TΨC loop of tRNA-Val (Fig. 2E). Furthermore, we uncovered a previously unreported Ψ site in position 8 of tRNA-Tyr (Fig. 2E). This is unlikely to be a false positive since the sequence motif (GUUC) is similar to that found in TΨC loops[13,15] (Fig. 2E).

Lastly, we obtained four different PUS knockout strains that are incapable of pseudouridylating specific positions in tRNAs and grown under optimal conditions (37 °C). We found Δ*truA* strains were unable to deposit Ψ in the anticodon-stem loop (Fig. 2F). Similarly, we did not detect Ψs in TΨC-loop and D-stem loop of tRNAs in Δ*truB* and Δ*truD* strains, respectively (Fig. 2F, Supplementary Data 2). These observations are consistent with established target locations of TruA, TruB, and TruD PUS enzymes[18,19]. However, although TruC modifies position 65 in the TΨC-loop, we still observed Ψ at this location in Δ*truC* mutant (Fig. 2F). One possible explanation could be that TruB may deposit Ψ in TruC sites since they recognize similar motif as described below. Thus, TruC and TruB may have redundant roles in this context. Altogether,

our approach accurately identifies known Ψ positions in rRNA and tRNAs of *E. coli*.

## A comprehensive Ψ landscape in *E. coli* transcriptome

In a previous study to map the repertoire of Ψ in *E. coli* RNAs, Schaening-Burgos et al. identified Ψ in only 42 mRNAs[11] using Pseudo-seq. However, using the highly sensitive BS-based approach, we detected high-confidence Ψ sites in 1217 mRNAs, which is 29 times above previous estimates (Fig. 3A, Supplementary Data 1). Most mRNAs only harbor a single Ψ site (Supplementary Fig. 2A). In addition, we found few mRNAs that are always pseudouridylated irrespective of growth conditions. For example, the multidrug efflux transporter, *mdtG*, contained Ψs in both normal and all tested stress conditions (Fig. 3B). In addition, almost 20% of detected pseudouridylated mRNAs are either involved in the biosynthesis of secondary metabolites or adaptation to diverse environments (Fig. 3C). This suggests a possible regulatory role of Ψ in response to stress, similarly to eukaryotes[5].

Next, we analyzed the motif frequency and distribution of all Ψ sites. The most frequent pseudouridylated motifs contained mostly G or C bases upstream of the Ψ site (Fig. 3D, Supplementary Data 3).

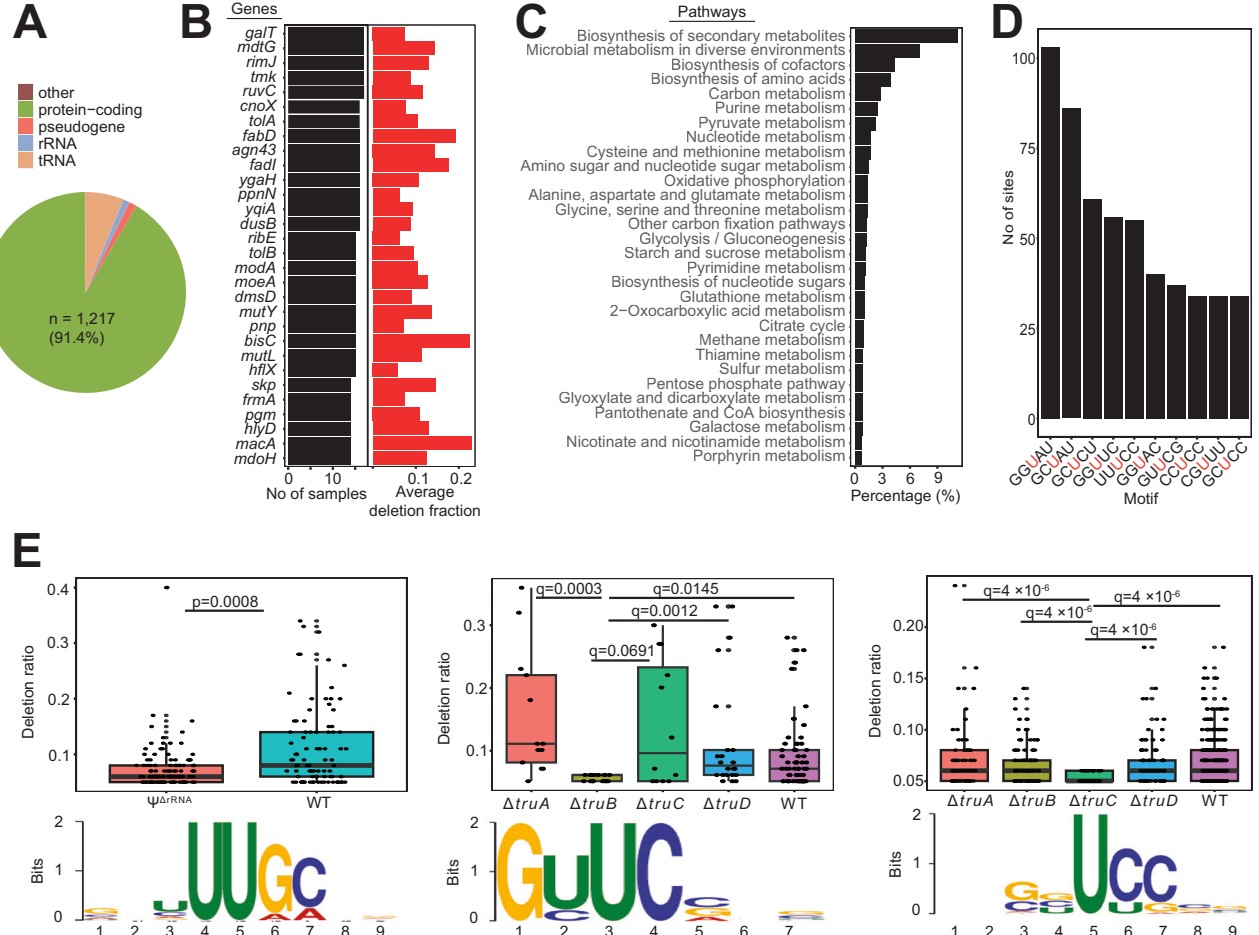

**Fig. 3 | Quantitative landscape of Ψ in E. coli transcriptome. A** Pie chart showing the distribution of Ψ across RNA types. **B** Ψ profiles of top 30 mRNAs. The barplot on the right shows the average deletion ratio across samples. For an mRNA with multiple Ψ positions, the sum of deletion ratios at all Ψ sites was calculated. **C** Pathway representation of mRNAs with Ψ. **D** Distribution of the top 10 sequence motifs for pseudouridylation. The red uridine residue represents the modification site. **E** Motifs with decreased deletion ratios in Ψ$^{\Delta rRNA}$ (left), Δ*truB* (middle), and Δ*truC* (right) strains, respectively. The image below shows the respective consensus recognition motifs. Significant differences between groups were computed with two-sided Wilcoxon rank sum test (left) or two-sided Wilcoxon rank sum tests with false discovery rate (FDR) adjusted *P* values (*q*). Center lines in boxplots represent the median and the edges represent the lower and upper quartiles. Whiskers show values that fall within 1.5× of the interquartile range. WT, $n = 90$; Ψ$^{\Delta rRNA}$ $n = 107$ (left panel). WT, $n = 64$; Δ*truA*, $n = 13$; Δ*truB*, $n = 17$; Δ*truC*, $n = 12$; Δ*truD*, $n = 22$ (middle panel). WT, $n = 345$; Δ*truA*, $n = 64$; Δ*truB*, $n = 107$; Δ*truC*, $n = 52$; Δ*truD*, $n = 92$ (right panel). Source data are provided as a Source Data file.

Interestingly, the most abundant motif (GGUAU), which we detected in over 100 sites, is also reported as being a frequent Ψ motif in the human transcriptome[15]. To further understand the sequence preference for each PUS enzyme, we compared the Ψ proportion (i.e., deletion ratio) for every motif in WT and mutant strains. In other words, a motif with a significantly lower deletion rate in a PUS mutant is most likely a sequence preference. We established that target sites for rRNA PUS enzymes occur in a consensus sequence comprised of 'UUGC' (Fig. 3E), which agrees with previously reported RluA recognition motif[41]. Similarly, we observed 'GUUC' as the main recognition sequence for TruB, in strong agreement with the homologous TRUB1 targets in human and yeast[13,20] (Fig. 3E). TruC, on the other hand, deposits Ψ mostly in 'UCC' sequences and possibly recognizes the 'GUUC' sequence similarly to TruB (Fig. 3E). However, we did not identify a consensus sequence for TruA or TruD. This may suggest a lack of sequence preference or pseudouridylation by other PUS enzymes.

We also examined the predicted secondary structure of all target sites using RNAfold[21] and observed pseudouridylation occurs predominantly in unpaired uridine sites (Supplementary Fig. 2B). However, deletion rates were similar in both paired and unpaired sites

(Supplementary Fig. 2C). Although, using synthetic probes, bisulfite treatment was previously shown to have a modest structural dependency due to both chemical accessibility of bisulfite[15] and transcriptase preference[13,15]; however, all Ψ sites were identified albeit with varying deletion ratios suggesting secondary structures do not qualitatively impede Ψ detection. Overall, Ψs are widespread across *E. coli* mRNAs and are mostly deposited in loop or hairpin structures by rRNA and tRNA PUS enzymes.

## Ψ stabilizes bacterial mRNA

Because Ψs are strongly linked to stability of non-coding RNA[22,23], we examined their possible role in mRNA stability and gene expression. To begin, we retrieved transcripts containing Ψ that can be assigned unambiguously to a PUS enzyme. In this case, sites with significantly lower Ψ (i.e., deletion ratio) in a PUS mutant, relative to WT, were considered a recognition site for a given PUS. Since multiple PUS can deposit Ψ at different locations in a single transcript, we assigned those candidates to multiple PUS. Next, we determined the abundance levels of candidate mRNAs from read counts. We observed that mRNAs unable to be pseudouridylated in PUS mutants were significantly less abundant (Fig. 4A). This was true for all PUS targets, with the sole

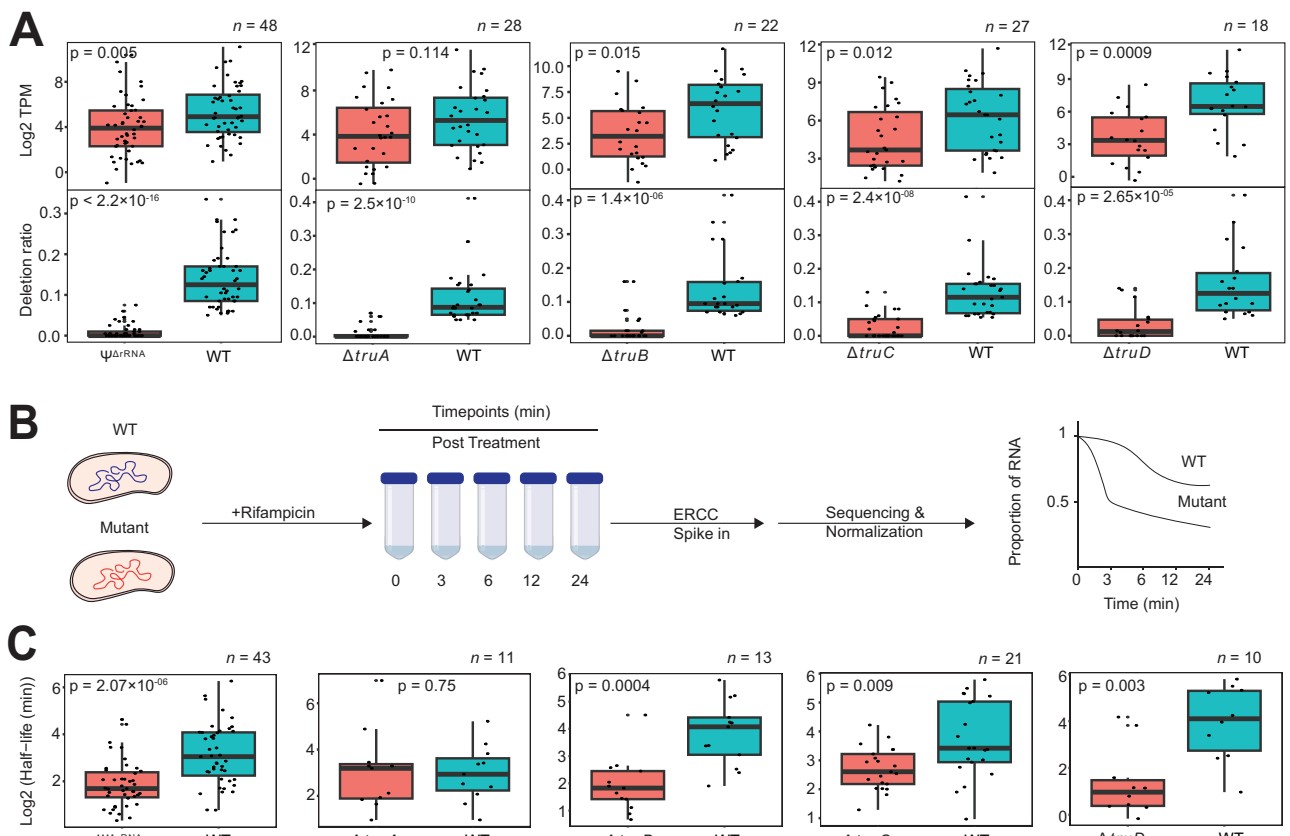

**Fig. 4 | Ψ is associated with mRNA abundance. A** Boxplot showing mRNA abundance (TPM) in WT and PUS mutants (top). Each datapoint represents the mean from 2 biological replicates. The lower boxplot shows the corresponding deletion ratio. Significant differences between groups were computed with two-sided Wilcoxon rank sum test. Center lines in boxplots represent the median and the edges represent the lower and upper quartiles. Whiskers show values that fall within 1.5× of the interquartile range. **B** Schematic representation of RIF-seq protocol to determine RNA half-lives. **C** Boxplot showing mRNA half-lives in WT and PUS mutants. The RNAs analyzed are identical to those in panel (**A**) but restricted to candidates whose half-lives could be determined. Significant differences between groups were computed with two-sided Wilcoxon rank sum test. Center lines in boxplots represent the median and the edges represent the lower and upper quartiles. Whiskers show values that fall within 1.5× of the interquartile range. Source data are provided as a Source Data file.

exception of TruA. In addition, we did not observe significant differences in mRNA abundance between WT and mutants when random control sets of equal size were compared, which suggests a link between Ψ and abundance (Supplementary Fig. 3A).

To further understand how Ψ may influence abundance, we performed rifampicin treatment and RNA sequencing (RIF-seq)[24] to measure mRNA decay rates (Fig. 4B). We included spike-in controls during sample preparation for normalization[25] and measured RNA abundance at 0, 3-, 6-, 12-, and 24-min post-transcription arrest with rifampicin. We calculated mRNA half-lives in both WT and PUS mutants and observed RNAs unambiguously modified by a PUS enzyme were significantly less stable in mutants except Δ*truA* (Fig. 4C). This suggests that the decrease in mRNA abundance is a consequence of transcript instability. We also compared the global RNA half-lives of all pseudouridylated and non-pseudouridylated transcripts and observed no significant differences in either the WT or mutant strains. (Supplementary Fig. 3B). Taken together, our results underscore a functional role of pseudouridylation in stabilizing bacterial mRNA.

### Quantitative mapping of Ψ landscape in human microbiome samples

To examine the ability of our approach to quantify pseudouridylation of mRNA in complex microbial communities, we recruited 18 subjects (13 healthy and 5 periodontitis patients) and retrieved oral plaque samples from the subgingival region (Fig. 5A). After BS treatment and

sequencing, we mapped reads to over 5000 genomes assembled from publicly available oral metagenomes[26] (Fig. 5A). Unlike our approach for *E. coli* isolates where we required a Ψ site to have a deletion rate <1% in untreated samples, we considered sites with higher deletions to accommodate indels arising from strain heterogeneity in complex microbial communities. However, we required the deletion rate in BS-treated samples to be greater than twofold in untreated samples. To further minimize false positives, we used a statistical approach that takes into consideration total read depth, deletion rate, and relationship between these parameters in BS-treated and untreated samples[14].

Next, we analyzed the microbiome composition and abundance using Kraken 2/Bracken[27]. We initially determined, using publicly available paired metagenomics and metatranscriptomics oral samples, that Kraken 2/Bracken accurately recapitulates microbial taxonomic information from metatranscriptome data (Supplementary Fig. 4). In our patient samples, the microbial community was dominated by diverse microbial species belonging to multiple phyla in agreement with the known microbial profiles of subgingival microbiomes[28,29] (Fig. 5B). In total, we identified 3534 Ψ sites from 3135 protein-coding transcripts, distributed across 218 species (Fig. 5C, D, Supplementary Data 4). Similarly to the *E. coli* data above, many of the annotated mRNAs harboring Ψ sites are either involved in the biosynthesis of antibiotics or other secondary metabolites, or adaptation to diverse environments (Fig. 5E). This suggests that a regulatory role of Ψ in response to stress may be widespread across bacteria.

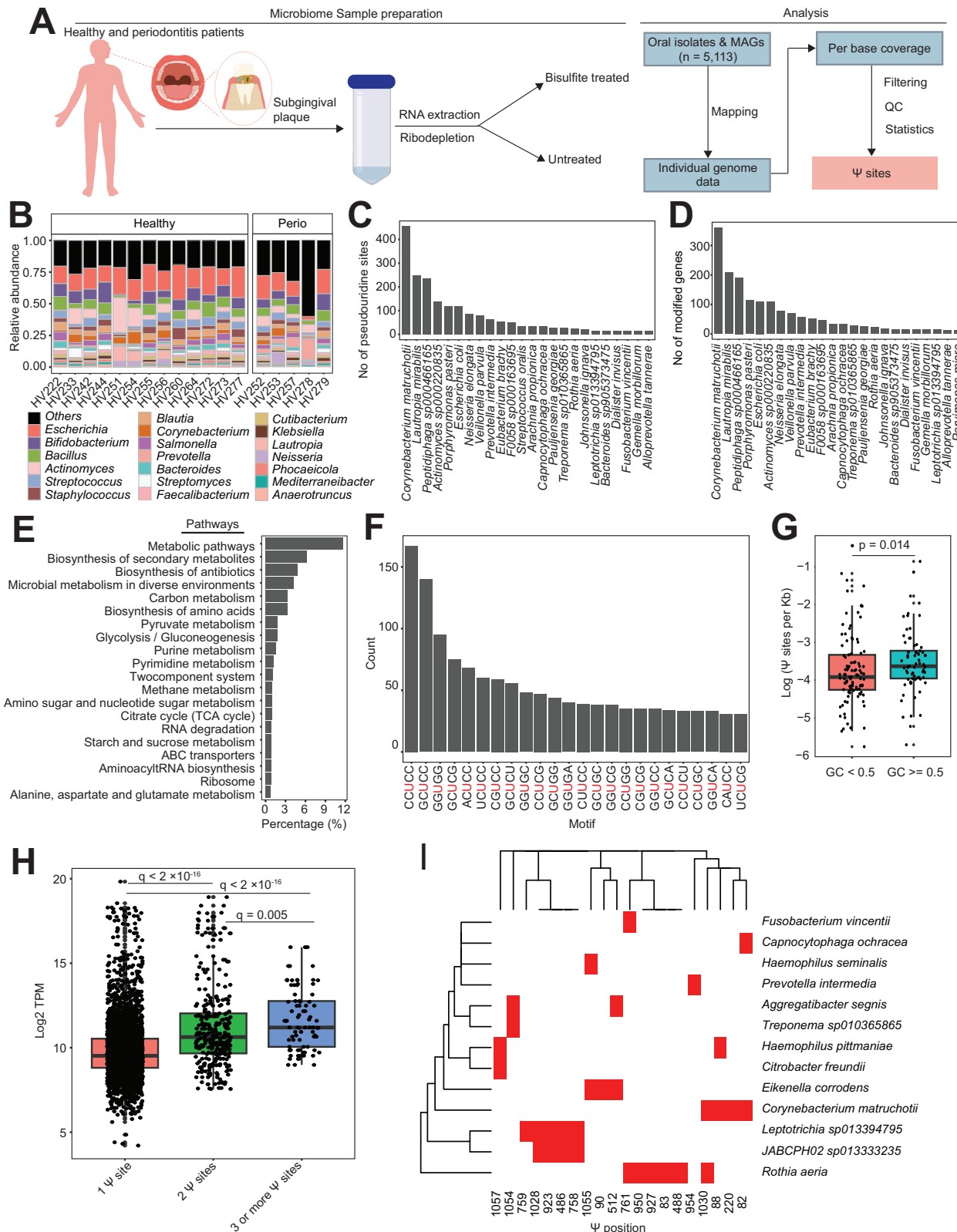

We also looked at the sequence preference for Ψ deposition and identified 254 different motifs — many of which were previously identified in *E. coli*. (Figs. 3D and 5F). Although we were unable to assign most motifs to a PUS enzyme, the top two motifs (CCUCC and GCUCC) resemble the TruC-associated motif in *E. coli* (Figs. 3E and 5F).

Because most of the enriched motifs are GC-rich, we hypothesized that pseudouridylation will be more predominant in mRNAs from GC-rich genomes. To exclude any potential sequence bias arising from chemical accessibility of bisulfite or transcriptase preference, we analyzed the number of identified Ψ sites per genome rather than the degree of pseudouridylation per site (i.e., deletion ratio). As

**Fig. 5 | Quantitative mapping of Ψ in the oral microbiome. A** Experimental and computational pipeline to process oral microbiome samples. **B** Microbial relative abundance from 18 subjects. **C** Barplot showing the top representative genome per genera harboring pseudouridine sites. **D** Barplot showing the number of pseudouridylated mRNAs in top representative species per genera. **E** Pathway representation of annotated mRNAs with Ψ. **F** Distribution of top sequence motifs for pseudouridylation. The red uridine residue represents the modification site. **G** Boxplot showing the normalized number of Ψ sites in mRNAs from genomes with GC content <0.5 (n = 101) and ≥0.5 (n = 69). The number of sites was normalized according to coverage depth and size. Significant differences between groups were computed with two-sided Wilcoxon rank sum test. Center lines in boxplots represent the median and the edges represent the lower and upper quartiles. Whiskers show values that fall within 1.5× of the interquartile range. **H** Boxplot showing the relationship between mRNA abundance and the number of Ψ sites per mRNA (1 site, n = 3,256; 2 sites, n = 322; ≥3 sites, n = 81). Significant differences between groups were computed with two-sided Brunner-Munzel tests with false discovery rate (FDR) adjusted P values (q). Center lines in boxplots represent the median and the edges represent the lower and upper quartiles. Whiskers show values that fall within 1.5× of the interquartile range. **I** Heatmap showing the high-confidence sites of Ψ in 16S rRNA from different bacteria (red = present; white = absent). Source data are provided as a Source Data file.

mentioned above, the motif sequence does not qualitatively impede Ψ detection[14]. For each genome, we performed normalization by taking into consideration the coverage depth (i.e., number of bases above the depth cutoff) and genome length. Interestingly, we observed significantly more Ψ sites in protein-coding transcripts from GC-rich genomes (Fig. 5G).

We also examined the relationship between Ψ and mRNA abundance in the community. Similarly, we focused on the number of Ψ sites observed in a transcript. Strikingly, the number of modification sites in an mRNA was positively correlated with transcript abundance (Fig. 5H). This data raises the possibility that multiple Ψ sites might be required to enhance the stability of an mRNA, though further investigation is necessary to confirm this requirement.

Finally, we sought to identify new Ψ sites in rRNA from our microbial community. The 16S of *E. coli*, for example, harbors a single Ψ site and the repertoire of rRNA modification sites across multiple bacteria have not been investigated. We began by retrieving 16S sequences from our genomes and only considered high-confidence sites to minimize false positives (i.e., sites identified in multiple genomes or samples) (Supplementary Data 5). We found new Ψ locations in 13 different species belonging to 5 phyla (Fig. 5I). Unlike *E. coli*, the 16S of some identified microbes had multiple modification sites. For instance, we identified 6 and 4 Ψ sites in the 16S RNA from *Rothia aeria* and *Corynebacterium matruchotii*, respectively. On the other hand, *Aggregatibacter segnis*, which belongs to the same Enterobacterales order like *E. coli*, harbors a high-confidence modification site (position 1054) in addition to the established *E. coli* site (position 512) (Fig. 5I, Fig. 2B). Modifications in similar sites (positions 1054–1057) were observed in other Proteobacteria species such as *Eikenella corrodens*, *Citrobacter freundii*, and *Haemophilus pittmaniae*.

Altogether, Ψ modifications are prevalent in microbial communities and our approach provided an avenue to study post-transcription regulation in microbiomes.

## Discussion

The lack of a suitable approach to study pseudouridylation in prokaryotes has made it difficult to identify exact modification sites in bacteria mRNAs. While a previous study could only identify modifications in 42 *E. coli* mRNAs[11] using Psedo-seq[12], our approach using a more sensitive BS-based method show that almost 30% of the *E. coli* transcriptome harbor Ψs. Moreover, our results provide a quantitative measure of Ψs. Therefore, our work represents a comprehensive analysis of bacterial mRNA pseudouridylation under defined conditions.

Although it is well established that Ψ stabilize rRNAs and tRNAs, there has been conflicting information regarding their role in mRNA stability. In one study, no association was found between Ψ levels and mRNA abundance in human HEK293T cells[15], whereas Dai et al. observed a significant association with mRNA levels[13]. In contrast, Nakamoto et al. found modifications by TruA homolog (PUS1) resulted in transcript instability in *Toxoplasma gondii*[30]. Nevertheless, our data suggest Ψ deposition is significantly associated with mRNA levels in *E. coli*.

In addition, there are currently no studies investigating post-transcription regulation in the microbiome. This is important because many pathogens use post-transcription regulation to control the expression of key virulent genes to adapt to their environment[31]. Likewise, we demonstrate that our approach can be extended to interrogate pseudouridylation in complex microbial communities. Similarly to our *E. coli* findings, we observed a correlation between Ψ sites and mRNA levels in the oral microbiota. As a result, the association between Ψ and transcript abundance may be a widespread phenomenon in prokaryotes.

Since modifications occur at uridine residues, it is reasonable to expect a higher frequency of Ψ sites in transcripts from AT-rich genomes. However, the reverse was observed from our data. While there is a minor sequence dependency using BID-seq protocol, this is unlikely to have under-detected Ψ sites in these genomes because a recently published Ψ detection protocol, which does not show bias for specific sequence motifs, also identified GC-rich sequences as main targets for PUS enzymes[32]. Furthermore, in a recent study using pure bacteria isolates, pseudouridines were detected in twice as many RNAs from *Pseudomonas syringae* (59% GC content) compared to *Bacillus cereus* (35% GC content) despite both strains having similar gene counts[33]. Nevertheless, more experiments are needed to validate our in-silico observations.

Recently, new Ψ sites for 16S rRNA were proposed for *P. aeruginosa*[33]. In our work, we identified new 16S modification sites in microbes belonging to 5 phyla. The presence of hypervariable regions in 16S sequences may partly contribute to the differences in both Ψ positions and number of sites detected across bacterial species.

In conclusion, our work provides a quantitative landscape of Ψ in *E. coli* and describes the interrogation of Ψ in microbiota samples. A major limitation of our approach is the high coverage requirement to call a Ψ position. This is evident in our inability to detect more Ψ sites in the microbiome samples. Similarly, when the Ψ site is next to multiple consecutive uridines, it is computationally challenging to determine the exact pseudouridylation site. Nonetheless, our findings have demonstrated the value of BS-based methods in the study of this important modification in bacteria. Future work on Ψ will not only expand our basic understanding of post-transcription regulation but also provide new insights into the role of pseudouridylation on protein translation landscape.

## Methods

### Bacterial strains and growth conditions

Unless stated otherwise, wild-type (MC415), Ψ$^{\Delta rRNA}$ (MC452), Δ*truA*, Δ*truB*, Δ*truC*, and Δ*truD* mutants were routinely grown on Luria-Bertani (LB) media at 37 °C (Supplementary Data 6). When required, 25 µg/ml of kanamycin was added to the media of Δ*tru* mutants. To induce stress conditions, bacteria (MC415 and MC452) were grown in the presence of ampicillin (0.4 µg/µl), gentamicin (0.4 µg/µl), or high salinity stress conditions (4% NaCl). In the latter, cells were cultured in Tryptic soy broth (TSB).

## RNA extraction, bisulfite treatment and library preparation

All fresh cultures were grown until $OD_{600}$ of 0.3 and subsequently preserved in RNA protect (Qiagen, #76506). Next, RNA was extracted from the cultures using RNeasy mini kit (Qiagen, #74104) and residual DNA was removed by DNA-free™ DNA Removal Kit (Thermo Fisher Scientific, #AM1906). After RNA quantification and quality analysis by Qubit Flex and Bioanalyzer, RNA samples were diluted with nuclease-free water to give ~400–700 ng in 22 μl of volume, which was thereafter split in two halves (i.e., 'BS-treated' and 'untreated' halves). Ribosomal RNA was depleted in both halves using Ribo-Zero Plus rRNA Depletion Kit (Illumina, #20040525) followed by fragmentation of all samples by addition of 0.9 μl of fragmentation reagents (Thermo Fisher Scientific, #AM8740). The mix was incubated at 95 °C for 20 s and fragmentation was stopped using 0.9 μl of stop reagent. Samples were immediately placed on ice.

Fresh bisulfite reagent (BSR) was prepared by adding 0.27 g of sodium sulfite (Sigma Aldrich, #901916) and 0.034 g of sodium bisulfite (Sigma Aldrich, #799394) to 900 μl of DEPC-treated water. Next, 45 μl of freshly prepared BSR was added to the 11 μl of fragmented 'BS-treated' half and incubated at 70 °C for 3 h. After incubation, 75 μl of nuclease-free water was added to the mix followed by 270 μl of RNA binding buffer (RNA Clean and Concentrator-5 column kit, Zymo Research, #R1015) and 400 μl of 100% ethanol. The entire mixture (~800 μl) was loaded onto the column and centrifuged for 30 s. The column was then washed with 200 μl of RNA wash buffer and subjected to desulfonation by adding 200 μl of RNA desulphonation buffer (Zymo Research #R5001-3-40) to the column. This was subsequently incubated at room temperature for 75 min. The column was washed using RNA wash buffer and eluted in 10 μl of elution buffer.

For the other 'untreated' half, samples were purified after fragmentation and eluted with 10 μl of elution buffer. Both purified BS-treated and untreated RNA samples were then annealed to random hexamers (50 μM) and subjected to cDNA first-strand synthesis using SuperScript™ IV reverse transcriptase (Thermo Fisher Scientific, #18090010). The reaction parameters were: 23 °C for 10 min, 50 °C for 1 h, and 80 °C for 10 min. Next, the second strand of cDNA was synthesized by DNA polymerase 1 (Thermo Fisher, #EP0041) in a reaction mixture containing 0.8 μl of RNAse H (Thermo Fisher, #EN0202) followed by incubation at 15 °C for 2 h and thereafter 75 °C for 10 min. This double-stranded cDNA was purified by adding 90 μl of Ampure beads (Beckman Coulter, #A63881) using exact instructions provided by Illumina (Illumina Stranded Total RNA Prep, Ligation with Ribo-Zero Plus reference guide). Subsequent steps for 3' ends adenylation, anchor ligation and indexing were performed using the Illumina Stranded Total RNA Prep, Ligation with Ribo-Zero (Illumina, #20040525), anchor plate (Illumina, #20040899), and DNA/RNA UD indexes set (Illumina, #20091646), respectively. Generated libraries were evaluated using Bioanalyzer and sequenced on the NovaSeq platform.

## Sample preparation for half-life estimation

Fresh bacterial cultures were grown in LB media to an $OD_{600}$ of 0.3. Transcription initiation was halted using rifampicin (0.5 mg/ml) and samples were collected at 0, 3, 6, 12 and 24 min post rifampicin treatment. The reaction was immediately stopped using stop solution (95% ethanol +5% Qiazol lysis solution (Qiagen, #79306)) followed by snap freezing in liquid nitrogen. The samples were allowed to thaw on ice, centrifuged, and bacterial pellets were resuspended in TE buffer containing 15 mg/ml lysozyme and 20 μl/ml proteinase K. To this mixture, 2 μl of a 1/10 of ERCC RNA spike-in sequences[25] (Invitrogen, #4456740) was added for downstream normalization. These spiked mixtures were further processed for RNA extraction and library preparation according to Illumina protocol.

## Human oral sample collection

Collection of human samples was performed on an IRB clinical protocol approved at the National Institutes of Health (NIH) Clinical Center (ClinicalTrials.gov ID NCT01568697). This study included 18 participants: 13 healthy volunteers and 5 patients with chronic periodontitis. All study participants provided written informed consent for participation in this study. Participants were deemed systemically healthy based on detailed medical history and select laboratory work up. In addition to systemic screening, periodontal status was assessed through detailed clinical oral evaluation. Subgingival plaque samples (tooth adherent biofilm) were removed using a Gracey Curette (HuFriedy Group) from patients and samples were immediately placed in DNA/RNA Shield Stabilization Solution (Zymo Research, #R1100-50).

## RNA extraction from human samples

Samples were vortexed to dissolve pellets and then added to 2 ml ZR BashingBead Lysis Tubes (0.1 & 0.5 mm) (Zymo Research, #S6012-50) with 600 μl of Qiazol lysis solution (Qiagen, #79306). Bead beating was performed for 5 min in FastPrep 24 homogenizer at maximum speed. After centrifugation, 180 μl of chloroform was added to the supernatant and centrifuged for another 15 min at 4 °C for phase separation. The aqueous phase was transferred to a new tube and 1.5:1 v/v 100% ethanol was added. The mixture was mixed by pipetting and RNA was extracted. Ribodepletion was performed on the extracted RNA using Illumina® Ribo-Zero Plus rRNA Microbiome depletion kit (Illumina, #20072062) and subsequent steps were processed as described in the previous section.

## Bioinformatics pipeline to detect Ψ

Adapter removal and quality trimming of raw reads were performed using Trim Galore[34]. We mapped clean reads to *E. coli* BW25113 genome (NZ_CP009273.1) using bwa-mem (v0.7.17)[35] and realigned reads using ABRA2 to improve detection of indels in downstream analysis[16]. Next, bam-readcount (v1.0.1) was used to retrieve nucleotide coverage and sequence variant information[36] and the resulting output was subsequently parsed using brc-parser.py[37]. To be considered a Ψ site, a uridine position needs to meet the following criteria: (i) coverage depth ≥20 in both BS-treated and untreated samples; (ii) deletion count ≥5 in BS-treated samples; and (iii) deletion ratio ≥5% in BS-treated samples but less than 1% in untreated samples. Ψ was derived as the difference in deletion ratio between BS-treated and untreated sites and a Ψ site was considered as high confidence if it was identified in at least 2 samples in the main dataset (Supplementary Fig. 1A).

## Motif analysis

To determine the sequence preference for each PUS, we retrieved flanking sequences of mRNA sites having deletion ratio >6%, which is the median deletion ratio in our dataset. We further excluded low-occurrence motifs which were identified less than 6 times. For every motif passing the threshold, deletion ratios were compared between WT and $Ψ^{ΔrRNA}$ strains, or between the four different *tru* mutants. A significantly decreased motif ($P$ value < 0.05, two-sided Wilcoxon rank sum test) in a mutant was considered a sequence preference for that PUS. To identify consensus sequences, motifs were submitted to MEME and the parameter to search the reverse complement strand was disabled[38].

## Pathway analysis

The pathways of genes associated with pseudouridylation were analyzed using KEGG. The KEGG Mapper search tool was employed to map and assign these genes to the corresponding pathways[39].

## RNA structure analysis

tRNA structures were predicted with RNAfold web server using default parameters[21]. To investigate base-pairing preference for pseudouridylation, 12-mer sequences flanking each side of the Ψ residue were used as input in the command-line version of RNAfold (-p -T 37 parameters).

## Estimation of mRNA abundance from RNA-seq and influence of Ψ on stability

Because untreated samples are equivalent to traditional RNA-seq, we quantified the mRNA abundance from read counts in untreated samples. To begin, the generated.bam files from previous mapping step were used as input in featureCounts (v2.1.1)[40] to retrieve unique read counts mapped to each gene. We included a pseudocount of one to each gene and normalized using 'transcript per million' (TPM).

To evaluate the role of Ψ on mRNA abundance, we initially calculated the Ψ-strength for each transcript[13]. Ψ-strength is defined as the sum of deletion ratios at all Ψ sites within one RNA. The gene-level deletion ratios were compared between WT and PUS mutants using only samples obtained at 37 °C because *tru* mutants were solely grown at optimum temperature. For a gene to be considered pseudouridylated by a given PUS, the difference in gene-level Ψ-strength between WT and a mutant must be ≥0.05 in both biological replicates.

To measure mRNA half-lives, reads from samples collected at timepoints post rifampicin treatment were mapped to *E. coli* genome plus the ERCC spike-in RNA sequences[25] and read counts per gene determined using featureCounts (v2.1.1)[40]. We used the 25 most abundant ERCC spike-in RNA for normalization and all timepoints were scaled relative to timepoint 0. Here, a normalization constant ($k$) was determined for each ERCC spike-in RNA using $k = \frac{ERCC_n}{ERCC_0} / \frac{Total_n}{Total_0}$, where $ERCC_n$ and $ERCC_O$ are RNA read counts at timepoints $n$ and $O$, respectively, while $Total_n$ and $Total_O$ are the total library size in samples $T_n$ and $T_O$[41]. The geometric mean of all $k$ values (at each timepoint), called normalization factor, was used to normalize the library. This was achieved by dividing raw read counts by the normalization factor. Transcripts per million (TPM) for each library were subsequently calculated. To account for the initial stability period observed post-rifampicin treatment, mRNA decay was modeled using a delayed first-order exponential function[42]. This two-phase model assumes a constant mRNA abundance during an initial lag phase, followed by exponential decay. Model fits for selected candidates were manually inspected to ensure data fidelity. Candidates with poor model fit were excluded.

To identify transcripts that are not PUS substrates (i.e., non-pseudouridylated), we filtered for those with, at least, moderate expression across all samples (TPM > 20) but no detectable Ψ in any sample. Requiring consistent expression ensured these high-confidence candidates were unlikely to be false negatives resulting from insufficient sequencing coverage in our pseudouridine detection pipeline.

## Microbiome analysis

We downloaded 5113 oral isolates and metagenome-assembled genomes (MAGs) from the Human Reference Oral Microbiome (HROM) database[26]. For each genome, we predicted and annotated genes using Prokka (v. 1.14.6)[43]. To reduce the computational load associated with analyses of a large dataset, we used a coverage cutoff to determine microbes for downstream analysis. For each subject, only genomes with >20X coverage in both BS-treated and untreated samples were processed. Reads were first mapped to the human genome to exclude non-microbial reads. The filtered reads were subsequently mapped to HROM genomes using bwa-mem (v0.7.17)[35], realigned using ABRA2[16], and per-base coverage was retrieved using bam-readcount (v1.0.1)[36]. Uridine positions passing the following filters were considered as pseudouridylated: (i) read depth ≥20 in both BS-treated and untreated

samples; (ii) deletion count ≥5 in BS-treated samples; (iii) deletion ratio ≥0.02 (i.e., 2%) in BS-treated samples; (iv) deletion ratio in BS-treated samples is, at least, 2 fold above deletion ratio in untreated samples; and (v) the $P$-value from Fisher's exact test should be <0.01 when total number of deletions and reads depth in BS-treated and untreated samples are compared. For sites passing the above threshold, the proportion of Ψ was derived as the difference in deletion ratio between BS-treated and untreated sites (i.e., Δ deletion ratio).

Data regarding the GC content of each genome was derived from Cha et al.[26]. To estimate the number of pseudouridine sites per microbe, we initially determined the total number of base positions (i.e., A, T, G, or C) with depth >20× in both BS-treated and untreated samples. This number was divided by 1000 to derive bases per Kb. The total number of Ψ sites was subsequently divided by the derived bases/Kb factor.

For 16S analyses, we used strict threshold cutoff to minimize false positive discoveries. First, we only processed Ψ sites with Δ deletion ratio >0.1 (i.e., 10%). Second, we required putative candidates to be present in at least 3 genomes or less than 3 genomes but identified in 3 or more samples. Finally, to evaluate the accuracy of Kraken 2/Bracken in metatranscriptomics data, we retrieved paired metagenomics and metatranscriptomics oral samples from Belstrøm et al.[44].

## Reporting summary

Further information on research design is available in the Nature Portfolio Reporting Summary linked to this article.

## Data availability

Raw sequence reads generated in this study were deposited in Sequence Reads Archive (SRA) under BioProject accession number PRJNA1414121. The oral isolates and metagenome-assembled genomes (MAGs) were downloaded from the Human Reference Oral Microbiome (HROM) database (https://www.decodebiome.org/HROM/listdir.php?directory=data/genome_catalog). For paired oral metagenomics and metatranscriptomics analysis (BioProject PRJNA396840 [https://www.ncbi.nlm.nih.gov/bioproject/396840]), the following SRA accessions were used; SRR5892221, SRR5892220, SRR5892219, SRR5892218, SRR5892225, SRR5892224, SRR5892223, SRR5892222, SRR5892227, SRR5892226, SRR5892199, SRR5892198, SRR5892197, SRR5892196, SRR5892203, SRR5892202, SRR5892201, SRR5892206, SRR5892194, SRR5892193. All other necessary data are included in the Supplementary Information. Source data are provided with this paper.

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

## Acknowledgements

We are grateful to NIDCR/NIDCD Genomics and Computational Biology Core (ZIC DC000086) for providing sequencing support and Dr. Michael O'Connor for sharing *E. coli* strains MC415 (WT) and MC452 ($\Psi^{\Delta rRNA}$). We also thank Laurie Brenchley and Teresa Wild for obtaining and processing subgingival plaque samples. This research was supported by the Intramural Research Program of the National Institutes of Health (NIH). The contributions of the NIH authors were made as part of their official duties as NIH federal employees, are in compliance with agency policy requirements, and are considered works of the United States Government. However, the findings and conclusions presented in this paper are those of the authors and do not necessarily reflect the views of the NIH or the U.S. Department of Health and Human Services.

## Author contributions

S.S. and A.E. conceived the study. S.S., B.W., B.Y., and M.P. performed wet-lab experiments. A.E. and N.D. conducted bioinformatics analyses. S.S., A.E., and N.M. analyzed the data. S.S. and A.E. wrote the manuscript. A.E. supervised the study.

## Funding

## Competing interests

The authors declare no competing interests.
