## [Transparent Peer Review file · Nature Communications]

Quantitative mapping of pseudouridines in bacterial RNA

Corresponding Author: Dr Akintunde Emiola

Version 0:

Reviewer comments:

Reviewer #1

(Remarks to the Author)

This paper presents a well-developed approach to studying pseudouridine (pseU) modifications in bacteria using a bisulfite (BS)-based method, providing valuable data on pseU sites in rRNA and mRNA. The authors effectively highlight the novelty of reporting the landscape of pseU sites in low-abundance RNA species, such as mRNA. The bisulfite reaction employed mirrors that used in PRAISE and BID-seq, and the application of this technology to bacterial samples, where polyA enrichment of mRNA is challenging, is commendable. This well-established protocol will undoubtedly benefit future research on bacterial pseU and other mRNA modifications.

However, considering the recent publications in Nature Communications that highlight significant discoveries and comprehensive studies on the biological functions of RNA modifications in various systems, including mammals and plants, this paper is not suitable for the journal. It has not adequately addressed the actual function of pseudouridine (pseU) in regulating bacterial gene expression through a comprehensive study. Here are a few suggestions for improvement:

1. In Fig. 2, the description of tRNA pseU could be more precise. The reverse transcriptase used in this study cannot read through heavily modified or methylated regions within bacterial tRNA. It would be beneficial to conduct an AlkB-based demethylation assay first to remove methylated bases, followed by library construction. This demethylation treatment could be applied immediately after RNA fragmentation, allowing the reverse transcriptase to read through all tRNA regions with accurate coverage, thereby ensuring reliable detection of pseU.
2. While the results showing approximately 1,200 pseU sites on bacterial mRNA in Fig. 2 appear solid, additional evidence is needed to elucidate the functional roles of these pseU modifications. Conducting mRNA stability assays or translation assays (such as Ribo-seq) would help clarify how mRNA pseU regulates mRNA stability and translation. Additionally, gene knockdown or knockout of pseU writer proteins could serve as a valuable perturbation to assess changes in mRNA pseU levels. Site mutations at pseU-modified U sites could also be employed to validate the effects of pseU on mRNA stability and translation.
3. In Fig. 6, considering the complexity and dynamic nature of mRNA expression in clinical samples, it would be prudent for the authors to monitor mRNA profiles before discussing pseU modifications. For example, if a pseU-modified mRNA from a specific bacterium uniquely decays in one patient while remaining intact in other patient samples, it would be essential to consider these mRNA expression dynamics in the discussion of pseU modification changes.
4. There is a minor point regarding the frequently used term "deletion fraction," which may cause confusion. Typically, the term "deletion ratio" is used to describe the observed reverse transcriptase deletion signature. Through calibration curves, these deletion ratios can then be converted into estimated pseU modification fractions. The pseU modification fraction is commonly referred to as pseU modification stoichiometry or pseU stoichiometry.

Reviewer #2

(Remarks to the Author)

The manuscript titled "Quantitative mapping of pseudouridines in bacterial RNA" by Sharma et al. describes the results of bisulfite-based deep sequencing in *E. coli* and oral microbiome samples to identify RNA pseudouridylation. The manuscript presents a stronger section describing the methodology and overall results up to line 166 (Figures 1-3), but the subsequent discussion on potential Ψ -dependent RNA stabilization, novel transcripts, asRNAs, sRNAs, and the oral

microbiome is substantially weaker.

Many of the claims rely on assumptions, and alternative explanations have not been sufficiently explored. The manuscript would benefit from clearer, more rigorously supported arguments. For example, the assertion that "Pseudouridine mapping enabled the investigation of differential expression of very low abundance transcripts associated with the general stress response" lacks sufficient supporting data to be considered valid.

In conclusion, I do not recommend the manuscript for publication in its current form.

Major points:

- Lines 86-91: Calculation of Ψ Proportion

- o A primary concern is the lack of consideration for the effect of secondary structures. The authors should clarify whether bisulfite treatment has a structural dependency similar to SHAPE reagents. Later (lines 163-166), a weak structure dependency of pseudouridylation is reported. Do the authors have data to demonstrate whether this is due to an enzymatic preference of the PUSs or the chemical accessibility of bisulfite? This should be discussed in more detail.

- o Even though the observed effect is relatively small, the background deletion rate should be subtracted before calculating the proportion. Additionally, data from bisulfite-treated PUS deletion strains should be considered as a background control to normalize for Ψ -independent effects of bisulfite treatment.

- The statement "One possible explanation could be low efficiency of TruC in vivo since it is known to modify only two tRNAs" does not logically explain the observed phenomenon. It should be clarified whether the authors are proposing that TruC is not relevant in this context or that TruC and TruD may have redundant roles.

- Figure 3E:

- o Why is the wild-type deletion fraction not shown for GUUC and UCC? Based on the boxplot medians, it appears that the deletion rate in the Δ TruC strain is also lower, suggesting that the GUUC motif is used by both TruC and TruD. However, there is a higher variance. This point should be explicitly discussed.

- Ψ Stabilization of Bacterial mRNA

- o The observed reduction in normalized read counts of selected RNAs in PUS deletion strains needs stronger validation.

- o A potential pleiotropic effect of PUS deletion should be ruled out. The authors should analyze random control sets of equal size for comparison.

- o Replicates appear to be averaged; a proper differential expression analysis using DESeq or a similar tool would strengthen the claim.

- o Even if the effect is statistically significant, a change in transcript stability is only one possible explanation. Indirect effects on RNA synthesis rates must be considered.

- o To conclude that Ψ increases stability, global half-life measurements are necessary. Additionally, even if a change in half-life is observed in a specific RNA, it could be an indirect and Ψ -independent effect of PUS deletion. Depending on the individual RNA, stability changes could be also either positive or negative.

- Ψ Profiling and Detection of Low Abundance Transcripts

- o The claim that Ψ profiling identifies differentially expressed transcripts better than conventional RNA-seq relies on the assumption that pseudouridylation rates are dependent on cellular RNA concentrations. However, this assumption is not supported by data in the manuscript.

- o Pseudouridylation may be influenced by co-factors, PUS concentrations, RNA-binding proteins, other RNAs, or other unknown factors.

- o Among the 14 genes with differential pseudouridylation, only 7 showed differential expression in RT-qPCR, which challenges the underlying hypothesis.

- o The manuscript should clearly distinguish between increased sensitivity for differential expression analysis and increased sensitivity for RNA detection in general.

- o The detection of new asRNAs and sRNAs is not inherently dependent on the bisulfite method, rather it results from different experimental conditions. Also e.g. an increased sequencing depth alone could enhance detection sensitivity. This should be explicitly stated throughout the manuscript.

- sRNA Functional Analysis

- o The discussion of sRNA function is superficial and does not substantially contribute to the main findings.

- o TargetRNA3 was trained on interactome data and may not be the most suitable tool for detecting functional sRNA interactions. Other tools, such as IntaRNA, might be more appropriate.

- o To confirm that sucA is a genuine sRNA target, compensatory mutations in both the target and sRNA should be tested to rule out secondary effects.

- Microbiome Analysis

- o The microbiome section could serve as a proof of principle, but given that *E. coli* alone has 1,331 Ψ sites, the detection of only 174 sites across the microbiome suggests that a significant fraction remains undetected. This indicates that the sensitivity of the approach needs substantial improvement before strong conclusions can be drawn.

- o The statement that TruB is the major PUS enzyme in the microbiome is weakly supported and should be rephrased. A more precise statement would be that a majority of the limited number of detected sites resemble the TruB-associated motif in *E. coli*.

- o The authors should also consider that the motif might be recognized by other enzymes in different bacterial species and that TruB may have additional or alternative motifs in other bacteria. Additionally, the observed enrichment of TruB sites appears weak and should be discussed accordingly.

Minor points:

- The abstract prominently states that bisulfite sequencing eliminates the need for strand-specific RNA sequencing of asRNAs. However, this is not a major advantage. In contrast, the detection of low-expressed unmodified asRNAs will be hindered without stranded sequencing.

- The sentence "Therefore, new approaches are required to evaluate the widespread distribution of Ψ in bacterial mRNAs"

assumes a priori that Ψ is widely distributed. It should be revised to say “evaluate the assumed widespread distribution of Ψ ” or “evaluate the widespread distribution of Ψ detected in this study.”

- The sentence “We began by extracting total RNA and enriching for mRNA through ribodepletion” is imprecise. The method enriches for all RNA species except ribosomal RNA, not just mRNA.
- Line 162: The structural analysis appears to be based on predictions. This and the used tool should be clearly stated in the text.
- Line 199: The sentence “Altogether, Ψ profiling can facilitate the discovery of low-expressed genes associated with bacterial adaptation to varying environmental conditions which may be difficult to study using conventional RNA-seq analysis pipelines” should be refined. Even if the underlying hypothesis is correct, the claim should be limited to facilitating the discovery of differential expression, rather than gene discovery in general.
- If a computational sRNA target prediction tool such as TargetRNA3 is used to generate data discussed in the main text, the tool should also be explicitly mentioned in the main text.

Version 1:

Reviewer comments:

Reviewer #1

(Remarks to the Author)

The manuscript has been significantly improved, and I support its publication in Nature Communications. The authors should further review the main text, legends, and data figures to ensure accuracy and correctness, after AIP.

Reviewer #2

(Remarks to the Author)

The authors addressed the points mentioned in the reviewer comments and significantly improved the manuscript.

However, the required and important inclusion of the global RNA stability data raised a new issue which needs to be addressed prior to acceptance.

The method to calculate the RNA stability based on reference 41 seems not to be the state of the art. It was shown that rifampicin decay curves show a characteristic “delay” of the exponential decline of the RNA dependent on the distance to the transcriptional start site. Not considering this delay leads to a false half-life which is often overestimated (PMID: 25583150, PMID: 37454177).

Please also compare the global RNA stabilities (all transcripts without pseudouridine) vs pseudouridine RNAs in WT and the mutant strains to look for pleiotropic patterns. How do the non-modified RNAs react in the mutants?

Assuming steady state, what proportion of the foldchange in the transcript abundance is explained by the stability differences for your selected candidates?

Minor observation:

Groups in Fig 5h seem to have unequal variances and sample sizes which violates the assumptions of the Wilcoxon test. The Brunner-Munzel might be more appropriate. The largely different sample size also hampers the general interpretations about stability and abundance of multiple modified RNAs.

Response to reviewers

We thank the reviewers for their insightful comments. This has significantly helped to improve our manuscript. Based on their comments, we restructured our manuscript to focus on the functional role of pseudouridines in bacterial mRNA. In this case, we excluded weaker sections such as those pertaining to sRNA analysis and detection of low abundance transcripts. In addition, we included 18 microbiome samples and new sequence data to measure mRNA global half-life in *E. coli*. Please find below our responses to the comments.

Reviewer 1

This paper presents a well-developed approach to studying pseudouridine (pseU) modifications in bacteria using a bisulfite (BS)-based method, providing valuable data on pseU sites in rRNA and mRNA. The authors effectively highlight the novelty of reporting the landscape of pseU sites in low-abundance RNA species, such as mRNA. The bisulfite reaction employed mirrors that used in PRAISE and BID-seq, and the application of this technology to bacterial samples, where polyA enrichment of mRNA is challenging, is commendable. This well-established protocol will undoubtedly benefit future research on bacterial pseU and other mRNA modifications.

We thank the reviewer for the positive feedback on our work

However, considering the recent publications in Nature Communications that highlight significant discoveries and comprehensive studies on the biological functions of RNA modifications in various systems, including mammals and plants, this paper is not suitable for the journal. It has not adequately addressed the actual function of pseudouridine (pseU) in regulating bacterial gene expression through a comprehensive study. Here are a few suggestions for improvement:

1. In Fig. 2, the description of tRNA pseU could be more precise. The reverse transcriptase used in this study cannot read through heavily modified or methylated regions within bacterial tRNA. It would be beneficial to conduct an AlkB-based demethylation assay first to remove methylated bases, followed by library construction. This demethylation treatment could be applied immediately after RNA fragmentation, allowing the reverse transcriptase to read through all tRNA regions with accurate coverage, thereby ensuring reliable detection of pseU.

We agree with the reviewer that a demethylation step may improve detection. However, there are already known pseudouridine sites in *E. coli* tRNAs. In our data, we identified **all these sites. In addition, our tRNA data was simply for validation purposes rather than discovery of new tRNA sites.**

2. While the results showing approximately 1,200 pseU sites on bacterial mRNA in Fig. 2 appear solid, additional evidence is needed to elucidate the functional roles of these pseU modifications. Conducting mRNA stability assays or translation assays (such as Ribo-seq) would help clarify how mRNA pseU regulates mRNA stability and translation. Additionally, gene knockdown or knockout of pseU writer proteins could serve as a valuable perturbation to assess changes in mRNA pseU levels. Site mutations at pseU-modified U sites could also be employed to validate the effects of pseU on mRNA stability and translation.

We totally agree. We performed RIF-seq experiments to measure global *E. coli* mRNA half-lives in both wild-type and PUS mutants. We observed that pseudouridine significantly enhances mRNA stability (Fig. 4).

3. In Fig. 6, considering the complexity and dynamic nature of mRNA expression in clinical samples, it would be prudent for the authors to monitor mRNA profiles before discussing pseU modifications. For example, if a pseU-modified mRNA from a specific bacterium uniquely decays in one patient while remaining intact in other patient samples, it would be essential to consider these mRNA expression dynamics in the discussion of pseU modification changes

We agree. Rather than using gene clusters (which are derived from multiple bacteria), we studied mRNA dynamics in individual microbes present in the community. Our new analysis is similar to that for *E. coli* isolates, but in this case, extended to multiple bacteria simultaneously (Fig. 5).

4. There is a minor point regarding the frequently used term "deletion fraction," which may cause confusion. Typically, the term "deletion ratio" is used to describe the observed reverse transcriptase deletion signature. Through calibration curves, these deletion ratios can then be converted into estimated pseU modification fractions. The pseU modification fraction is commonly referred to as pseU modification stoichiometry or pseU stoichiometry.

We agree and have implemented the change throughout the manuscript.

Reviewer 2

The manuscript presents a stronger section describing the methodology and overall results up to line 166 (Figures 1-3), but the subsequent discussion on potential Ψ -dependent RNA stabilization, novel transcripts, asRNAs, sRNAs, and the oral microbiome is substantially weaker.

Many of the claims rely on assumptions, and alternative explanations have not been sufficiently explored. The manuscript would benefit from clearer, more rigorously supported arguments. For example, the assertion that "Pseudouridine mapping enabled the investigation of differential expression of very low abundance transcripts associated with the general stress response" lacks sufficient supporting data to be considered valid.

We agree with the reviewer. We have excluded weaker sections on sRNAs and low abundance transcripts and refocused our manuscript on the functional role of pseudouridines. In addition, we increased the number of microbiome samples analyzed from 2 to 18 which has substantially improved our analysis.

Major points:

- Lines 86-91: Calculation of Ψ Proportion

A primary concern is the lack of consideration for the effect of secondary structures. The authors should clarify whether bisulfite treatment has a structural dependency similar to SHAPE reagents. Later (lines 163-166), a weak structure dependency of pseudouridylation is reported. Do the

authors have data to demonstrate whether this is due to an enzymatic preference of the PUSs or the chemical accessibility of bisulfite? This should be discussed in more detail.

Bisulfite-based protocols such as BID-seq¹ and PRAISE² (which we adopted) do not specifically report on structural dependency of bisulfite treatment. However, we analyzed their data and made the following observations. (i) In Zhang et al¹., using synthetic probes containing different sequence motifs, bisulfite treatment was shown to have a very minor structural dependency. In one instance, deletion ratio for AGΨCU was 85% whereas AGΨUG was 50%. We suspect this is both due to chemical accessibility of bisulfite and transcriptase preference^{1,2}. Nevertheless, pseudouridines were robustly identified albeit with varying deletion ratios suggesting secondary structures do not qualitatively impede Ψ detection. We have included these points in the main manuscript text (lines 158 – 165).

o Even though the observed effect is relatively small, the background deletion rate should be subtracted before calculating the proportion. Additionally, data from bisulfite-treated PUS deletion strains should be considered as a background control to normalize for Ψ-independent effects of bisulfite treatment.

We agree. We have now subtracted background deletion rates prior to calculating deletion ratios. In addition, it is difficult to use PUS deletion strains to normalize for Ψ-independent effects because some sites are modified by more than one PUS enzyme (Lines 129 – 130).

The statement “One possible explanation could be low efficiency of TruC in vivo since it is known to modify only two tRNAs” does not logically explain the observed phenomenon. It should be clarified whether the authors are proposing that TruC is not relevant in this context or that TruC and TruB may have redundant roles.

We apologise for the lack of clarity. We have modified the sentence to suggest that TruC and TruB may have redundant roles since they recognize similar motifs (Lines 130 – 133).

Figure 3E:

o Why is the wild-type deletion fraction not shown for GUUC and UCC? Based on the boxplot medians, it appears that the deletion rate in the ΔtruC strain is also lower, suggesting that the GUUC motif is used by both TruC and TruD[B]. However, there is a higher variance. This point should be explicitly discussed.

We have included boxplots for the WT (Fig. 3E). We have also included the suggestion on the possibility of TruC and TruB recognizing the GUUC motif (Lines 154 -156).

Ψ Stabilization of Bacterial mRNA

o The observed reduction in normalized read counts of selected RNAs in PUS deletion strains needs stronger validation. A potential pleiotropic effect of PUS deletion should be ruled out. The authors should analyze random control sets of equal size for comparison.

We agree. We have analyzed random control sets of equal size which further corroborates a link between pseudouridines and mRNA abundance (Supplementary Fig. 3).

o Replicates appear to be averaged; a proper differential expression analysis using DESeq or a similar tool would strengthen the claim.

With DESeq or similar tools, it requires, at least, triplicate samples for robust analysis. In our case, we only had two biological replicates.

o Even if the effect is statistically significant, a change in transcript stability is only one possible explanation. Indirect effects on RNA synthesis rates must be considered.

We agree. We further performed RIF-seq to measure global RNA half-lives in WT and PUS mutants. Our data indicate pseudouridine stabilizes *E. coli* mRNA (Fig. 4).

o To conclude that Ψ increases stability, global half-life measurements are necessary. Additionally, even if a change in half-life is observed in a specific RNA, it could be an indirect and Ψ -independent effect of PUS deletion. Depending on the individual RNA, stability changes could be also either positive or negative.

We agree. Please see response above

- Ψ Profiling and Detection of Low Abundance Transcripts

o The claim that Ψ profiling identifies differentially expressed transcripts better than conventional RNA-seq relies on the assumption that pseudouridylation rates are dependent on cellular RNA concentrations. However, this assumption is not supported by data in the manuscript.

o Pseudouridylation may be influenced by co-factors, PUS concentrations, RNA-binding proteins, other RNAs, or other unknown factors.

o Among the 14 genes with differential pseudouridylation, only 7 showed differential expression in RT-qPCR, which challenges the underlying hypothesis.

o The manuscript should clearly distinguish between increased sensitivity for differential expression analysis and increased sensitivity for RNA detection in general.

o The detection of new asRNAs and sRNAs is not inherently dependent on the bisulfite method, rather it results from different experimental conditions. Also e.g. an increased sequencing depth alone could enhance detection sensitivity. This should be explicitly stated throughout the manuscript.

We totally agree. As mentioned above, we have excluded the section on detection sensitivity of low abundance from our manuscript.

- sRNA Functional Analysis

o The discussion of sRNA function is superficial and does not substantially contribute to the main findings.

o TargetRNA3 was trained on interactome data and may not be the most suitable tool for detecting functional sRNA interactions. Other tools, such as IntaRNA, might be more appropriate.

o To confirm that *sucA* is a genuine sRNA target, compensatory mutations in both the target and sRNA should be tested to rule out secondary effects.

We agree with the reviewer. We have excluded this section as mentioned above.

- Microbiome Analysis

- o The microbiome section could serve as a proof of principle, but given that *E. coli* alone has 1,331 Ψ sites, the detection of only 174 sites across the microbiome suggests that a significant fraction remains undetected. This indicates that the sensitivity of the approach needs substantial improvement before strong conclusions can be drawn.

We agree. To improve this section, we analyzed 18 microbiome samples (compared to 2 in our initial submission) and utilized a more efficient microbiome ribodepletion kit to enrich for non-rRNA. We identified 3,534 Ψ sites from 3,135 protein-coding transcripts, distributed across 218 species in the microbiome (Fig. 5).

- o The statement that TruB is the major PUS enzyme in the microbiome is weakly supported and should be rephrased. A more precise statement would be that a majority of the limited number of detected sites resemble the TruB-associated motif in *E. coli*.

- o The authors should also consider that the motif might be recognized by other enzymes in different bacterial species and that TruB may have additional or alternative motifs in other bacteria. Additionally, the observed enrichment of TruB sites appears weak and should be discussed accordingly.

We agree and have modified the text to exclude claims of TruB being the major PUS enzyme (Line 211).

Minor points:

- The abstract prominently states that bisulfite sequencing eliminates the need for strand-specific RNA sequencing of asRNAs. However, this is not a major advantage. In contrast, the detection of low-expressed unmodified asRNAs will be hindered without stranded sequencing.

We have excluded the section on asRNA from the manuscript.

- The sentence “Therefore, new approaches are required to evaluate the widespread distribution of Ψ in bacterial mRNAs” assumes a priori that Ψ is widely distributed. It should be revised to say “evaluate the assumed widespread distribution of Ψ ” or “evaluate the widespread distribution of Ψ detected in this study.”

We agree and have modified the text appropriately (Lines 54 -55).

- The sentence “We began by extracting total RNA and enriching for mRNA through ribodepletion” is imprecise. The method enriches for all RNA species except ribosomal RNA, not just mRNA.

We agree and have modified the sentence (Line 79).

- Line 162: The structural analysis appears to be based on predictions. This and the used tool should be clearly stated in the text.

This has been included in the main text (Line 158).

- Line 199: The sentence “Altogether, Ψ profiling can facilitate the discovery of low-expressed genes associated with bacterial adaptation to varying environmental conditions which may be

difficult to study using conventional RNA-seq analysis pipelines” should be refined. Even if the underlying hypothesis is correct, the claim should be limited to facilitating the discovery of differential expression, rather than gene discovery in general.

As mentioned above, we have excluded this section from the manuscript.

- If a computational sRNA target prediction tool such as TargetRNA3 is used to generate data discussed in the main text, the tool should also be explicitly mentioned in the main text.

Similarly, we have excluded this section from the manuscript.

References

1. Zhang, L.S. et al. BID-seq for transcriptome-wide quantitative sequencing of mRNA pseudouridine at base resolution. *Nat. Protoc.* **19**, 517-538 (2024).
2. Zhang, M. et al. Quantitative profiling of pseudouridylation landscape in the human transcriptome. *Nat. Chem. Biol.* **19**, 1185-1195 (2023).

Response to reviewers

We thank the reviewers once again for their positive comments. Please find below our responses to the remaining comments.

Reviewer 1

The manuscript has been significantly improved, and I support its publication in Nature Communications. The authors should further review the main text, legends, and data figures to ensure accuracy and correctness, after AIP.

We thank the reviewer for the positive feedback.

Reviewer 2

The method to calculate the RNA stability based on reference 41 seems not to be the state of the art. It was shown that rifampicin decay curves show a characteristic “delay” of the exponential decline of the RNA dependent on the distance to the transcriptional start site. Not considering this delay leads to a false half-life which is often overestimated (PMID: 25583150, PMID: 37454177).

We thank the reviewer for pointing us in this direction. We have now accounted for the initial stability period observed post-rifampicin treatment by modelling mRNA decay using a delayed first-order exponential function. This two-phase model assumes a constant mRNA abundance during an initial lag phase, followed by exponential decay. The new results are similar to previous ones, and the conclusions remain the same (Fig. 4C).

Please also compare the global RNA stabilities (all transcripts without pseudouridine) vs pseudouridine RNAs in WT and the mutant strains to look for pleiotropic patterns. How do the non-modified RNAs react in the mutants?

We compared the global RNA half-lives of all pseudouridylated and non-pseudouridylated transcripts and observed no significant differences in either the WT or mutant strains. (Supplementary Fig. 3B).

Assuming steady state, what proportion of the foldchange in the transcript abundance is explained by the stability differences for your selected candidates?

We are careful not to assign a specific numerical value to the contribution of half-lives to abundance, as such an estimate would likely be inaccurate given the complexity of bacterial gene expression. For instance, accelerated degradation may feedback to alter the transcription rate for specific genes.

Minor observation:

Groups in Fig 5h seem to have unequal variances and sample sizes which violates the assumptions of the Wilcoxon test. The Brunner-Munzel might be more appropriate. The largely

different sample size also hampers the general interpretations about stability and abundance of multiple modified RNAs.

We agree. We performed statistical analysis using Brunner-Munzel test and still observed significant differences between the groups (similarly to Wilcoxon test results) (Fig. 5H). Nevertheless, we have modified our interpretation to make it cautionary (Line 230).